# Chimerism Testing by Next Generation Sequencing for Detection of Engraftment and Early Disease Relapse in Allogeneic Hematopoietic Cell Transplantation and an Overview of NGS Chimerism Studies

**DOI:** 10.3390/ijms241411814

**Published:** 2023-07-23

**Authors:** Abdelhamid Liacini, Gaurav Tripathi, Amanda McCollick, Christopher Gravante, Peter Abdelmessieh, Yuliya Shestovska, Leena Mathew, Steven Geier

**Affiliations:** 1Immunogenetics Laboratory, Pathology and Laboratory Medicine, Temple University and Hospital, Lewis Katz School of Medicine, 3401 N. Broad St., Office B242, Philadelphia, PA 19140, USA; gaurav.tripathi@tuhs.temple.edu (G.T.); amanda.mccollick@tuhs.temple.edu (A.M.); christopher.gravante@tuhs.temple.edu (C.G.); leena.mathew@tuhs.temple.edu (L.M.); steven.geier@tuhs.temple.edu (S.G.); 2Fox Chase Cancer Center Medical Group, Temple Health, Philadelphia, PA 19140, USA; peter.abdelmessieh@fccc.edu (P.A.); yuliya.shestovska@tuhs.temple.edu (Y.S.)

**Keywords:** next-generation sequencing (NGS), chimerism, STR, allo-HCT, engraftment

## Abstract

Chimerism monitoring after allogenic Hematopoietic Cell Transplantation (allo-HCT) is critical to determine how well donor cells have engrafted and to detect relapse for early therapeutic intervention. The aim of this study was to establish and detect mixed chimerism and minimal residual disease using Next Generation Sequencing (NGS) testing for the evaluation of engraftment and the detection of early relapse after allo-HCT. Our secondary aim was to compare the data with the existing laboratory method based on Short Tandem Repeat (STR) analysis. One hundred and seventy-four DNA specimens from 46 individuals were assessed using a commercially available kit for NGS, AlloSeq HCT NGS (CareDx), and the STR-PCR assay. The sensitivity, precision, and quantitative accuracy of the assay were determined using artificially created chimeric constructs. The accuracy and linearity of the assays were evaluated in 46 post-transplant HCT samples consisting of 28 levels of mixed chimerism, which ranged from 0.3–99.7%. There was a 100% correlation between NGS and STR-PCR chimerism methods. In addition, 100% accuracy was attained for the two external proficiency testing surveys (ASHI EMO). The limit of detection or sensitivity of the NGS assay in artificially made chimerism mixtures was 0.3%. We conducted a review of all NGS chimerism studies published online, including ours, and concluded that NGS-based chimerism analysis using the AlloSeq HCT assay is a sensitive and accurate method for donor-recipient chimerism quantification and minimal residual disease relapse detection in patients after allo-HCT compared to STR-PCR assay.

## 1. Introduction

Hematopoietic cell transplant (HCT) therapy can be curative or provide the possibility of an extended quality of life for patients suffering from a variety of malignant and non-malignant hematological disorders, including lymphoma, leukemia, aplastic anemia, thalassemia, and congenital immune diseases. Usually, human leukocyte antigens (HLA) matched donor hematopoietic cells are used to transplant patients. Despite HLA matching, major barriers to HCT success include post-transplant complications such as infections, graft versus host disease, and disease relapse [1,2,3,4]. Among these complications, disease relapse is the leading cause of death [5].

Regularly monitoring patient/donor cell ratios, or chimerism, is critical for detecting changes in engraftment status, which can impact patient care. Post-transplant blood specimens from HCT recipients who attain complete remission status prior to transplantation generally should demonstrate complete chimerism, 95–100% donor-derived DNA. Maintenance of complete chimerism status indicates successful engraftment, whereas the presence of increasing recipient cells suggests a potential relapse [6]. Estimates of the percentage of donor chimerism are important in recipients at risk for malignancy relapse; it can initiate therapeutic interventions of early relapse or engraftment failure, such as with donor lymphocyte infusions and chemotherapy [7]. 

To estimate donor chimerism, a number of methods are available which use polymorphic markers to differentiate between donor and recipient cells, including qPCR, Fluorescent in situ hybridization, variable number of tandem repeats (VNTRs), short tandem repeats (STRs), and single nucleotide polymorphism (SNP) based assessment [8,9,10,11,12,13,14]. Except for identical twins, these tests are useful in all donor-recipient combinations, HLA, and disease types [8,9,10,11,12,13,14]. Despite their advantages, these assays have several drawbacks. For instance, the qPCR assays are inaccurate when the donor’s or recipient’s smaller population is large, typically require several single plex or low-level multiplex reactions to be set up for each patient, and STR-based assays are limited when few informative markers are available for early relapse detection, and limited sensitivity at low levels of chimerism (>5%) [15]. The introduction of next-generation sequencing (NGS) technologies has advanced the field of transplant biology. The NGS chimerism panels are generally concordant with STR-based panels, but they have greater sensitivity and specificity [14,16]. NGS-based chimerism assays use single nucleotide polymorphisms (SNPs) and provide standardized workflow with automated data analyses.

In the present study, we validated and evaluated chimerism monitoring using the commercially available NGS test CareDx AlloSeq HCT (www.caredx.com/alloseq-hct, accessed on 19 October 2021) kit for engraftment monitoring (Appendix A). We compared the NGS results to our existing STR-based chimerism detection assay AmpFLSTR™ Identifiler™ Plus panel, a commercially available multiplex assay from Applied Biosystems. We evaluated sensitivity, precision, and quantitative accuracy using pre-transplant DNA, simulated chimerism constructs [17,18], artificially created chimeric specimens, proficiency testing specimens, and a blind challenge of chimeric specimens. We also conducted a detailed comparative overview of all published papers on chimerism testing, having NGS as a main molecular biology technique for chimerism detection.

## 2. Results

Post-HCT chimerism analysis has been a critical diagnostic tool for the determination of engraftment and relapse. Due to their sensitivity, specificity, and rapid throughput, NGS platforms are advancing the field of medical genomics. This study was to establish and validate the NGS-based chimerism analysis and compare it to our existing PCR-STR-based assay for early detection of relapse/minimal residual disease in allo-HCT. 

### 2.1. Accuracy and Proficiency Testing

Accuracy was defined as the closeness of a measured value to the true value, where we included 20 different samples for accuracy measurement. NGS AlloSeq HCT accuracy in artificially mixed chimerism was measured by comparing the expected percentage mixed DNA with the observed percentage chimerism in artificially mixed gDNA samples extracted from fresh whole blood. We tested 36 post-transplant HCT samples, consisting of 28 levels of mixed chimerism, which ranged from 0.03–99.7% (Figure 1a). A very high coefficient of correlation was observed (R^2^ > 0.99) for samples containing a single or two donors (Figure 1a). Other chimerism NGS testing reports, also observed coefficient correlations of >0.99% for monitoring of mixed chimerism in HCT [19,20].

The accuracy of the CareDx NGS, AlloSeq HCT assay was also investigated via the ASHI Engraftment Monitoring (EMO) program. As a part of the ASHI EMO program, which included five unknown samples per survey, we participated in two surveys comprising a total of 10 unknown DNA specimens that were analyzed (observed value) and compared to the mean scores from all other participating labs (expected value). The percentage donor chimerism for all 10 unknown DNA fell within the range of expected mean value ±2SD and received 100% ‘GOOD’ results. Results for testing five representative ASHI 2022 EMO-1 surveys (test samples EMO-156, EMO-157, EMO-158, EMO-159, and EMO-160) are shown in Figure 2a, which demonstrate a high level of assay accuracy and quality (Figure 2a).

### 2.2. Comparison of NGS Chimerism with STR-PCR

We evaluated the performance of an NGS-based chimerism assay and compared it to the existing laboratory method based on Short Tandem Repeat (STR) analysis. Chimerism testing was compared between NGS (observed donor chimerism) and STR-PCR (expected donor chimerism) by testing 46 blinded samples obtained from the Children’s Hospital of Philadelphia (CHOP). Informative STR markers were obtained to compare the results from NGS CareDx. 

The results showed a high correlation between expected STR and observed chimerism levels by NGS, with a correlation of ≥0.99 (Figure 1b). However, the NGS assay was more precise and accurate compared to the STR-PCR results. The STR-PCR assay did not produce reliable results at mixtures below 5%; NGS was more sensitive and had significantly better precision at mixed chimerism levels >0.3% (Figure 1b). NGS also produces an additional number of informative markers in the same samples; STR-PCR has a limited number of informative markers (Table 1). The average number of informative markers using NGS AlloSeq HCT was 110 (Figure 2b). NGS data was more reliable, allowing enough depth read (>500) for each informative marker to be included.

### 2.3. Performance Specification of NGS Chimerism

The STR and NGS assay performance was compared for the different DNA input amounts in samples. The results of NGS (observed donor chimerism) with STR (expected donor chimerism) showed a 100% correlation between expected STR and observed chimerism levels of NGS chimerism with a 100% concordance (Figure 1).

### 2.4. Analytical Sensitivity of NGS

The assay must be very sensitive to detect the disease relapse in allo-HCT from a fraction of the target DNA. The sensitivity of the STR-PCR multiplex panel assays has previously been reported as 5% [10,17,21,27,28,29,30]. For calculation of the analytical performance of the NGS assay, which includes the limit of blank (LOB) and limit of detection (LOD), was tested (Figure 3a,b). 

The limit of blank is defined as the 95th percentile of the fraction of the average measured background in blank samples and was determined to be 0.02–0.06% (average 0.04%). The assays’ sensitivity or ability to detect the lowest detectable level (LOD) of chimerism was evaluated. Artificially mixed samples with varying ranges of low chimerism (0.1–1%) were analyzed. The sensitivity of the NGS assay for the artificially mixed chimerism was 0.3%. The limit of detection experiments demonstrated a 0.3% sensitivity, as shown in Figure 3b, when analyzing three mixtures at 0.1% with NGS. Blank represents the average noise of blanks (0.04%) (Figure 3b). The average of all triplicates originating from the same series was calculated (Figure 3c). The slope was 1.1407. The y-intercept was 0.1112, and the R^2^-value was 0.99. These results produced a linear and accurate assay performance with minimal systematic bias. 

### 2.5. Precision and Reproducibility Testing

Inter-run reproducibility measures precision across multiple runs of the assay. The mean, SD (ranges from 0.0–1.6%), and overall CV for repeatability or measurement of inter-run variation was 5.08% and ranged from 0 to 11.21%, demonstrating excellent harmonization of results obtained (Figure 4a). For inter-run variability following, 0.1%, 0.2%, 0.3%, 0.4%, 0.5%, 1%, 5%, and 10% donor chimerism levels were tested for a total of three replicates of each. The CV was 5.08% (range 0–11.21%) (Figure 4b).

### 2.6. Inter-Tech Variability

The results of inter-run variability among technicians for all 12 samples with varying % donor chimerism (100%, 99.9%, 99.8%, 99.7%, 99.6%, 99.5%, 90.0%, 80.0%, 60.0%, 40.0%, 20.0%, and 0.0%) showed CV < 4% (Figure 4c). There was excellent concordance between the runs with an SD < 1% in 9 samples and <2% in 3 samples of donor-mixed chimerism, and an overall %CV was less than 4%. The observed value was obtained from the CHOP given for the STR analysis as a blinded sample for comparison with NGS chimerism results. The results showed 100% concordance of the observed and expected % donor chimerism in inter-tech variability.

## 3. Discussion

Chimerism testing is important for engraftment monitoring and detecting impending relapse in patients to assist physicians in making immunotherapy decisions such as DLI and second HCT planning [31]. The most commonly used methods for monitoring engraftment in laboratories are STR-PCR and quantitative PCR (qPCR) [8,9,10,11,12,13,14]. This study validated and compared the NGS AlloSeq (CareDx) and STR-PCR results for the systematic monitoring of chimerism in allo-HCT patients. The NGS assay was assessed for its accuracy, analytical sensitivity, specificity, precision, and reproducibility. The AlloSeq NGS assay demonstrated 100% concordance with STR-PCR results, but it is more sensitive, which makes it suitable for chimerism testing. The results produced a high correlation of ≥0.99 between expected (STR-PCR) and observed chimerism levels (NGS AlloSeq) (Figure 1b). 

It is crucial that the assay be sensitive enough to detect low levels of mixed donor chimerism, especially in patients with malignant diseases. STR-PCR assays displayed a sensitivity of 2–5% or above (Table 1). The sensitivity of the NGS assay in this study was 0.3% compared to the STR-PCR assay sensitivity of 5.0%. The NGS assay was able to detect mixed chimerism with a low amount of DNA (10 ng). In case of <10% donor-mixed chimerism and early post-HCT stages, the NGS assay is able to detect donor-mixed chimerism, and it is better for detecting minimal residual disease [11,27]. Real-time PCR-based determination of mixed chimerism exhibited higher sensitivity and better accuracy <5 CV% at >20% mixed chimerism but limited sensitivity at >2.5% [8,11,15]. In contrast, the reported sensitivity of qPCR methods (0.1%) is higher than STR-PCR; however, the accuracy of qPCR (coefficient of variation 30–50%) is much lower than STR (2–5%), especially in the samples where donor-mixed chimerism <10% [11,32]. Due to this limitation, many clinical laboratories use both methods (STR and qPCR) to detect mixed chimerism [19], which is costly and labor-intensive. To validate the NGS chimerism assay, a variety of sample types were used to determine if NGS AlloSeq matched STR-PCR approaches. We performed serial dilution of the DNA of two unrelated healthy volunteers in ratios ranging from 0.1–10%, with a total of three replicates for each sample. The R2-value was 0.99 (Figure 3 and Figure 4), and the sensitivity was 0.3% for the NGS assay in artificially mixed chimerism (Figure 3b). The accuracy of the assay was investigated by participation in a proficiency testing program, the ASHI Engraftment Monitoring program. The results for all 10 unknown DNA were in the range of expected mean value ±2SD and received 100% ‘Accurate’ results. Matthew et al. [18] presented data on using a FORENSIC NGS chimerism platform to measure mixed chimerism and reported a sensitivity of 1%. They reported a significant correlation between NGS and STR-PCR chimerism methods with 100% accuracy in the proficiency testing program.

Next-generation sequencing has made it possible to create a variety of applications, including NGS’s ability to quantify chimerism. In contrast to PCR approaches, which employ a panel of polymorphisms (SNPs or INDELs), which is sometimes limited to one or two informative markers, the AlloSeq HCT provides us with a panel of 202 SNPs to monitor allograft dysfunction and chimerism testing. We conducted a comparative overview of all chimerism studies that utilized NGS as the primary molecular biology technique for chimerism detection after allo-HCT (Table 1). To the best of our knowledge, the presented study is the only one testing the AlloSeq HCT NGS assay in the setting of chimerism assessment post-allo-HCT from the USA, Pennsylvania, Philadelphia. The comparison points focused on ethnicity, technology, sensitivity, specificity, the number of informative markers, and comparison with STR-PCR or qPCR as one of the gold standards (Table 1). The NGS assay demonstrated greater sensitivity (ranges from 0.01–1.0%) and specificity (100%) than the STR-PCR assay, excellent multiplexing capability with a broad selection of informative markers (Table 1), and applicability for nearly all populations [32,33]. The results of this overview signify that although STR-PCR is the most commonly used method, other methods, such as qPCR, dPCR, and NGS, have been developed to overcome the technical limitations of STR-PCR [34]. Alizadeh et al. [11] established qPCR-based chimerism testing for monitoring donor cell engraftment in allo-HCT recipients and selected 19 specific sequence polymorphisms belonging to 11 human biallelic loci located on 9 different chromosomes. They reported a sensitivity of 0.1%, which is comparable to NGS and higher than the STR-PCR assay. In spite of the high accuracy and precision, it was challenging to achieve a precision higher than 40%. We used the AlloSeq CareDx Illumina NGS platform to overcome the sensitivity and specificity issue by STR and qPCR assays. There are many potential advantages of NGS over qPCR and STR-PCR, including high throughput, low cost per target, ability to work in multiplex, 24 h turnaround time from processing to reporting, and simultaneous detection of large numbers of informative markers (current paper, Table 1). These advantages provide the opportunity to detect chimerism at an early stage with a small amount of DNA. The NGS assay, with a wide range of SNP analysis and higher sensitivity, enables testing of allo-HCT from sibling donors [35]. All studies/laboratories that adopted and validated chimerism detection using NGS displayed minimal technical artifacts relative to the STR assay, as shown in Table 1. STR assay suffers from numerous artifacts, such as stutter peak, peak height imbalance, allele dropouts, and dye interference. Two major NGS platforms were used in all papers for NGS-based chimerism detection: Ion Torrent and Illumina (Table 1). Kim et al. [14] and Aloisio et al. [16] used Ion Torrent, and others used Illumina. The sensitivity and specificity of assays across both platforms are comparable (Table 1). In the study by Aloisio et al. [16], a customized panel of 44 amplicons was used for NGS-based chimerism quantification. They designed a bioinformatics tool for genotyping and quantification of NGS data, which provides clinicians with a novel tool for chimerism testing following allo-HCT. Li et al. [22] evaluated chimerism and microchimerism (when donor chimerism is in the micro range <1%) using the SNP-based NGS assay in forty-eight HLA-mismatched stem cell microtransplantations. They developed an improved SNP-NGS method with increased sensitivity (0.01–0.05%) and minimal DNA input (8–200 ng). They concluded that SNP-NGS can accurately detect the microchimerism status of donor cells early in patients with acute myeloid leukemia [22]. Vynck et al. [20] performed an assessment of the Devyser high-throughput-sequencing-based assay for chimerism monitoring after transplantation in allo-HCT and compared it with the PowerPlex 16 HS SRT-PCR assay (Table 1). There were 24 SNPs markers included in the NGS assay and 16 in the STR-PCR, but the NGS assay had 3% informative markers, and the STR-PCR assay did not have any (0%) [20]. There are studies that raise some concern regarding the use of NGS in chimerism testing. Kim et al. [14] used a total of 15 whole bone marrow samples collected from ten acute myeloid leukemia patients and reported a relative quantification analysis of SNP markers by NGS in one human bone marrow chimerism sample with a 4.9% chimerism percentage.

Recently, a paper published by Picard et al. [26] using a total of 38 samples, a comparison of chimerism quantification data for two new digital PCR systems and two NGS-based chimerism quantification methods was performed. They found that all three existing NGS kits, Devysr^®^ (Devyser Chimerism), CareDX (AlloSeq HCT), and GenDx (NGStrack) are similar in terms of analytical performance. They concluded that AlloSeq proposes an analysis in the absence of a contributor, and it is not necessary to perform genotyping for all contributors. In the case of AlloSeq HCT, due to the presence of a high number of markers, it avoids false-negative results caused by chromosomal deletions in the relapse of some malignancies. Chimerism detection using an NGS AlloSeq HCT kit is the easiest technique, with only one mix (vs. 7 for NGStrack) and one PCR (vs. 2 for Devyser chimerism NGS). However, they also concluded that in comparison with all new methods in clinical practice, digital PCR is faster (result within the day vs. 72 h), easier to use, and easier to interpret than that of all NGS methods, which enables earlier detection of relapses in allo-HCT. 

NGS has been adopted in many laboratories due to its innovative technology, but there are some limitations, such as high infrastructure costs, lack of trained technologists, bio-informatics facility, storage of enormous amounts of data, and relatively lengthy processing and analysis time. As a technical issue, NGS can also generate a high background error rate and recurrent amplification of identical reads [20]. Many scientists and clinicians have reported that long run time and the high cost are the two most significant barriers to the use of NGS technologies in the clinical field in comparison to STR-PCR methods [23,36]. There are other approaches to facilitate donor-recipient chimerism quantification and early relapse monitoring of minimal residual disease. Recent developments in single-cell spatial transcriptomic technology enable the examination of the spatial patterns of cell communication and hold promise for unraveling the intricate ligand-receptor interactions that occur across different cell types [37]. Triozzi et al. [38] suggested the use of blood bioenergetics and metabolomics as predictive biomarkers of patient response to immune checkpoint inhibitor therapy.

## 4. Materials and Methods

### 4.1. Sample Selection

Samples analyzed by NGS at Temple Immunogenetics Laboratory included archived clinical specimens with results by STR, n = 46, test results; n = 24 artificial mixtures of volunteer donors; and 10 ASHI external EMO proficiency specimens. Patients enrolled in the validation study were divided according to their disease diagnoses. We had the disease status for forty-three patients, out of which 55.8% of patients were diagnosed with Acute Myeloid Leukemia (AML = 24), 16.3% Myelodysplastic Syndromes (MDS = 7), and 7% of each of Acute Lymphocytic Leukemia (ALL = 3) and Hodgkins Lymphoma (HL = 3). Two patients were diagnosed with Chronic Myelomonocytic Leukemia (CMML = 2; 4.7%) and one each of Chronic Myelogenous Leukemia (CML = 1; 2.3%), Neuroblastoma (NBL = 1; 2.3%), Severe combined immunodeficiency (SCID = 1; 2.3%) and Telomere Biology Disorder (TBD = 1; 2.3%) (Table 2). The criteria for choosing the STR-PCR chimerism samples for comparison with NGS were dependent upon the presence of one or more informative markers. When the recipient and donor have no common allele, which is fully informative, or when the recipient and donor share one common allele, which is heterozygous but still informative [18]. 

### 4.2. DNA Extraction

The genomic DNA(gDNA) of the donors and recipients was extracted from the fresh whole blood collected in Acid Citrate Dextrose tubes (Becton, Dickinson and Company, Franklin Lakes, NJ, USA) according to the manufacturer’s protocol (Qiagen kit details). The quantity and quality of gDNA were quantified by the NanoDrop (ND-1000 spectrophotometer, NanoDrop Technologies, Wilmington, DE). The extracted DNA used in the study had absorbance ratios of A260/A280 > 1.8 and A260/A230 > 1.7. The extracted DNA samples were stored at −20 °C.

### 4.3. Artificial Mixed Chimeric Specimen Preparation

To simulate different levels of hemato-lymphatic chimerism observed after allogeneic HCT, twenty-four artificial mixtures of volunteer donors’ genomic DNA (gDNA) derived from mixed peripheral blood of related and unrelated donors at fractions ranging from 0.1–100% were mixed in different proportions, 100%, 95%, 90%, 80%, 70%, 60%, 50%, 40%, 30%, 20%, 10%, 5%, 2%, 1% and 0% of donor DNA and each dilution point was regarded as a unique sample. The artificial mixtures were combined into four separate pairs to simulate samples displaying mixed chimerism. Forty-six additional DNA samples were included from archived clinical DNA specimens with historical results by STR to be compared to NGS AlloSeq HCT testing.

### 4.4. NGS, Panel and Chimerism Analysis

Libraries were prepared according to the manufacturer’s protocol (CareDx AlloSeq-HCT). Reagents for the assay were included in the Kit. Each run was performed, including one positive and one negative control. It is a kit-based one-step multiplex assay. The assay requires 10 ng of gDNA input for the library preparation. In the first step, PCR amplification is performed using unique index primers; each sample requires one PCR reaction, followed by pooling and purification of samples. The pooled libraries are quantified and loaded onto the MiSeq sequencer (Illumina, San Diego, CA, USA) after denaturation (Appendix A). The rapid workflow allows for the analysis of 48 samples in less than 24 hrs (Appendix A). The low sample requirement facilitates chimerism assessment in multiple cell subsets. The AlloSeq HCT assay is a targeted next-generation sequencing assay that makes use of variations in single nucleotide polymorphisms (SNPs) to calculate the proportion of recipient- and donor-derived DNA that is present in a post-transplant sample. The sequencing reaction was carried out using the MiSeq v3 Reagent kit for 150 cycles. For the sequencing reaction, a customized sample sheet was created, and AlloSeq HCT Software was used to analyze the data from the fastq files. Amplification and indexing are combined into a single reaction by the special amplification procedure, which reduces human interaction and manipulation errors. For all the PCR programs and step-by-step procedures, we followed the instruction manual protocol. 

The NGS panel includes 202 SNPs present on all autosomal chromosomes (excluding XY chromosome) to differentiate between recipient and donor(s) gDNA. The 202 target SNPs are: (i) distributed across the 22 pairs of autosomal chromosomes in the human genome, (ii) biallelic, with an allele frequency between 0.4 and 0.6 across all populations, (iii) located in genomic regions that can be sequenced with high confidence, and (iv) not in linkage disequilibrium or associated with diseases. 

Sequence data is analyzed using AlloSeq HCT software, which outputs the percentage of DNA (% DNA) for up to three distinct genomes (genetic contributors) detected in post-transplant samples. The calculation of % DNA from recipient and donor(s) present in post-transplant samples is achieved by determining the fraction of different nucleotides sequenced at each SNP location evaluated. Recipient and donor(s) genotypes obtained using AlloSeq HCT are required to calculate the % DNA obtained from each genetic contributor present in the post-transplant sample. The number of reads categorized as “reference” (Ref) and “alternative” (Alt) of the Reference Sequence hg19 (human genome 19; Genome Reference Consortium) is used to calculate the variant allele frequency (VAF) for each marker. The analysis time for each sample is 30 s, and multiple samples can be analyzed on AlloSeq HCT software simultaneously. The software is user-friendly. It keeps track of the genotyping data and automatically detects if there is contamination or mix-ups between the samples. 

### 4.5. Accuracy, Performance Specification, and Analytical Sensitivity

Quantitative accuracy is defined as the closeness of an observed value to the true/known value. The accuracy of the NGS chimerism assay was investigated in artificial chimerism and simulated chimerism mixtures using NGS AlloSeq HCT. Twenty-four data points were generated. A comparison was made of NGS results versus the reference values of artificial mixtures. The quantitative accuracy in artificial mix chimerism results was calculated by comparing the expected % mixed DNA with the observed % chimerism. We also participated in an external proficiency program from ASHI (ASHI Proficiency Testing Engraftment Monitoring Program); as a part of the ASHI EMO program, we assessed 10 samples. Assessment of percentage (%) chimerism and quality of specimens were evaluated.

To compare the performance specification of NGS with an established technique such as STR analysis for chimerism detection, we tested 46 blinded samples received from the Children’s Hospital of Philadelphia (CHOP), Philadelphia, PA, USA. The received samples were blind to compare the results of NGS (observed donor chimerism) with STR (expected donor chimerism). 

The analytical sensitivity of the assay is defined as the smallest detectable amount of analyte that can be reliably distinguished from zero in the test system, also known as the detection limit. The sensitivity was determined using a statistical comparison of variation between samples with concentrations equal to the limit of quantification and samples with no analyte. To define the limit of detection or sensitivity of NGS assay in our cohort, mixed chimerisms of 0.1%, 0.2%, 0.3%, 0.4%, 0.5%, and 1.0% were artificially made. Three replicates of the mixed chimerism levels were tested. Precision was assessed within the run (intra-run reproducibility) using three samples, performed by three technicians, and between the runs (inter-run reproducibility) performed for all 12 samples with varying percentages of donor chimerism (100%, 99.9%, 99.8%, 99.7%, 99.6%, 99.5%, 90.0%, 80.0%, 60.0%, 40.0%, 20.0%, and 0.0%) by the three different technicians on three different days. The inter-tech variability was calculated by the following sets of samples using donor-mixed chimerism: 0.0%, 20%, 40%, 60%, 80%, 90%, 99.5%, 99.6%, 99.7%, 99.8%, and 99.9%. The run was performed by three different technicians on three different days (Figure 3c). 

To determine informative alleles for post-transplant surveillance, the STR chimerism data of recipient and donor pre-transplant samples were available. In NGS chimerism, the bi-allelic SNP approach of 1 reference allele and 1 variant allele with equal frequency maximizes the likelihood of identifying informative markers. The NGS assay was performed and validated at Temple University Hospital Immunogenetics Laboratory, Philadelphia, PA, USA.

### 4.6. Statistical Analyses

Figures were computed using GraphPad prism. All statistical analyses were performed using IBM SPSS Statistics software (SPSS Inc. v22.01, Chicago, IL, USA). Spearman’s coefficient correlation was calculated to compare the chimerism quantity with a proportion of DNA mixed in artificial chimeric DNA. The unpaired *T*-test or Kruskal–Wallis test was used to compare between groups. Mean, standard deviation (SD), and cumulative variance (CV) were calculated for different chimerism quantity comparisons. NGS and previous STR chimerism results were compared for concordance. A *p* value < 0.05 was considered statistically significant. 

## 5. Conclusions

AlloSeq NGS chimerism assay (CareDx) is an easily performed test that is informative for patients of all racial and ethnic backgrounds. The assay is fast, allowing analysis and reporting of up to 48 samples in less than 24 h on a flow cell (Appendix A). AlloSeq NGS assay exhibited higher sensitivity, precision, and accuracy in comparison to the STR-PCR assay. It offers enhanced diagnostic performance with a variety of clinical applications and can be an alternative for STR-PCR assays. The comparative overview of all chimerism studies based on NGS as the primary molecular biology technique for chimerism detection after allo-HCT demonstrated greater sensitivity and specificity than the STR-PCR assay. 

## Figures and Tables

**Figure 1 ijms-24-11814-f001:**
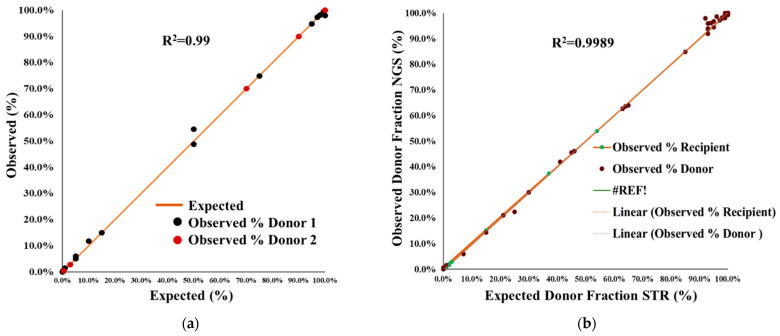
NGS concordance in the known (artificially mixed chimerism) and NGS concordance with STR in blind samples (**a**) Results of artificially mixed chimerism with two known donor DNA samples at varied levels of simulated artificial mixed chimerism. The results represent a good correlation of the expected chimerism in percent DNA fractions in samples with two unrelated genetic contributors. Signal linearity with the expected % DNA fraction was R^2^ = 0.99 across all donors. (**b**) Comparison of NGS concordance with STR in blind samples received from Children’s Hospital of Philadelphia. The results showed a high correlation (R^2^ > 0.99) between expected (STR) and observed chimerism levels (NGS).

**Figure 2 ijms-24-11814-f002:**
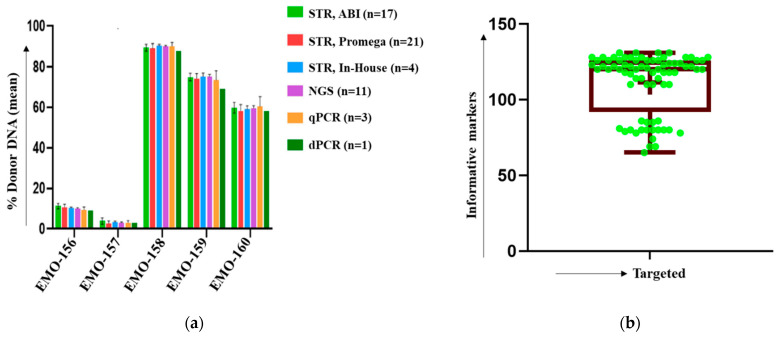
ASHI external proficiency specimens and average number of informative markers using NGS-based chimerism detection. (**a**) Results for testing five representative ASHI 2022 EMO-1 surveys (test samples EMO-156, EMO-157, EMO-158, EMO-159, EMO-160). There was 100% concordance in ASHI EMO results between results reported and other lab participants using different test platforms. (**b**) Average number (n = 110) of informative markers using NGS Chimerism assay. Green dots represent the informative markers.

**Figure 3 ijms-24-11814-f003:**
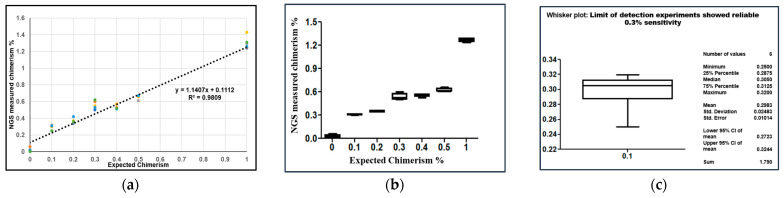
Sensitivity of the assay: The performance of the assay in artificially mixed samples with varying ranges of chimerism (0.1–1%) in triplicates. (**a**) The quantification linearity of the assay was determined. The average of all triplicates originating from the same series was calculated. The slope was 1.1407. The y-intercept was 0.1112, the R^2^-value was 0.99. These data display a linear and accurate performance of the assay with minimal systematic bias. (**b**) Boxplot (whisker plot) showing NGS assessment of Limit of Blank (LOB) and limit of detection (LOD) or sensitivity of micro-chimerism. (**c**) The assay’s ability to detect the lowest level (LOD) of mixed chimerism was evaluated. Artificially mixed samples with varying ranges of chimerism were analyzed and calculated. The sensitivity was 0.3%.

**Figure 4 ijms-24-11814-f004:**
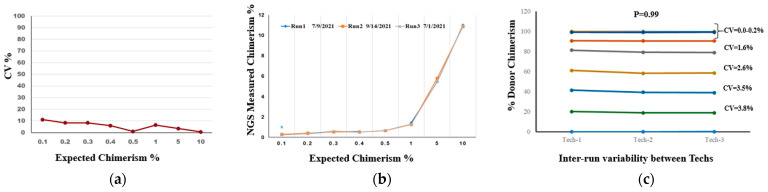
Reproducibility (Intra-run variability and Inter-run variability) between the runs and among the technicians. (**a**) Reproducibility (Inter-run): Data indicate that percent coefficients of variation (%CV) for chimerism. The average percent coefficient of variation (%CV) was 5.08 and ranged from 0 to 11.21%. (**b**) For repeatability or measurement of inter-run variation, an excellent harmonization of the obtained results. (**c**) Figure represents the results of Inter –run variability among the technicians performed for all 12 samples with varying % donor chimerism (100%, 99.9%, 99.8%, 99.7%, 99.6%, 99.5%, 90.0%, 80.0%, 60.0%, 40.0%, 20.0%, and 0.0%) showed CV < 1%.

**Table 1 ijms-24-11814-t001:** Details of NGS chimerism assay published in scientific journals and comparison with STR assay.

No.	Population	Sample Size	Number of Markers and Discription	Methodology	Sequencing Kit	Technology	NGS: Informative SNPs	NGS Sensitivity/Specificity	STR/NGS Analysis	STR: Informative SNPs	STR Sensitivity/Specificity	References
1	Republic of Korea	15 BM samples	124 SNPs; among these, 90 autosomal SNPs	NGS vs. STR	HID-Ion AmpliSeq™ Identity Panel (Life Technologies, Thermo Fisher Scientific, Waltham, MA, USA)	Ion PGM™ System (Life Technologies)	20.4 (13–32)	<1%/100%	11 STRs: GenomeLab Human STR Primer Set (Beckman Coulter, Fullerton, CA, USA)	5.7 (2–9)	1–5%	[14]
2	Italy	10	44-amplicon custom chimerism panel	NGS	Ion AmpliSeq custom chimerism (ACCh) panel	Ion Torrent	16	0.04–1.0%/100%	AmpFlSTR Identifiler Plus PCR Amplification Kit (Thermo Fisher Scientific, Inc., CA, USA)	not available	4%/100%	[16]
3	USA	not specified	N= 230: Primer A: Amelogenin, 27 autosomal STRs, 24 Y-STRs, 7 X-STRs, 94 informative SNPs. Primer Mix B, 78 aiSNPs	NGS vs. STR	NGS ForenSeq	MiSeq—FGx- Illumina	not available	1%/100%	CE-STR: 16–21 STR loci	not available	5%/99%	[18]
4	Belgium	422	24 loci with a known biallelic insertion/deletion polymorphism	NGS vs. STR	Devyser next-generation sequencing chimerism assay	MiSeq or NextSeq550Dx instrument (Illumina, San Diego, CA)	15	0.1%/100%	Power- Plex 16 HS PCR-CE assay (Promega assay)	not available	2–5%/100%	[20]
5	Sweden	651 samples	24 indels	NGS vs. STR and RQ-PCR	Devyser Chimerism NGS kit by Devyser AB, Stockholm, Sweden	Illumina MiSeq	9 (~40% informative)	0.1%/100%	In-house STR marker and RQ-PCR by Alizadeh et al. [21]	1 for STR and 2 for RQ-PCR	2–5%/100%	[19]
6	People’s Republic of China	48	48 SNPs	In house 48 primer sets	SNP-NGS	Illumina MiSeq	not available	0.01–0.05%/100%	STR	not available	1–10%	[22]
7	Republic of Korea	53	121 SNPs	Agilent Technologies, Santa Clara, CA, USA	Customized target kit	Illumina HiSeq4000 platform	(9–37)	0.5–1.0%/91.7%	AmpFlSTR Identifier PCR Amplification (Applied Biosystems, Warrington, UK)	25.5 (9–41)	1–5%	[23]
8	France	91	24 indels	NGS vs. STR and cdPCR	Devyser Chimerism NGS kit by Devyser AB, Stockholm, Sweden	Illumina MiSeq	not available	0.1%/100%	Crystal Digital PCR KMR kits (GenDX, Utrecht, The Netherlands)	not available	0.1%/100% for cdPCR	[24]
9	France	24	24 indels	NGS	Devyser Chimerism NGS kit by Devyser AB, Stockholm, Sweden	Illumina MiSeq	8	0.1%/100%	ddPCR	not available	0.10%	[25]
10	France	24	24 indels	NGS	Devyser^®^ panel (Devyser Chimerism NGS)	Illumina MiSeq	Not available	0.1%/100%	AlloSeq HCT (CareDx) and NGStrack	not available	0.3% for AlloSeq and 0.5% for NGStrac	[26]
11	Temple University Hospital, USA	174	202 SNPs across all autosomal chromosomes	NGS vs. STR	AlloSeq HCT (CareDx)	Illumina MiSeq	110	0.3%/100%	AmpFlSTR Identifiler Plus PCR Amplification kit (Thermo Fisher Scientific, Inc.)	2–9	5%/99%	Current Paper

**Table 2 ijms-24-11814-t002:** Distribution of patient disease diagnoses included for validation study.

Disease Diagnosis	N = 46	%
Acute Lymphocytic Leukemia (ALL)	3	7.0
Acute Myeloid Leukemia (AML)	24	55.8
Chronic Myelogenous Leukemia (CML)	1	2.3
Chronic Myelomonocytic Leukemia (CMML)	2	4.7
Hodgkins Lymphoma (HL)	3	7.0
Myelodysplastic Syndromes (MDS)	7	16.3
Neuroblastoma (NBL)	1	2.3
Severe combined immunodeficiency (SCID)	1	2.3
Telomere Biology Disorder (TBD)	1	2.3
Unknown	3	7.0

## Data Availability

Not applicable.

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
