# Peer review of "Chimerism Testing by Next Generation Sequencing for Detection of Engraftment and Early Disease Relapse in Allogeneic Hematopoietic Cell Transplantation and an Overview of NGS Chimerism Studies"

_ijms, 2023, doi:10.3390/ijms241411814_

Round 1
Reviewer 1 Report
Comments and Suggestions for Authors
The study compares the applicability of NGS-based chimerism monitoring to the gold-standard STR-PCR assay. Several performance parameters like accuracy, sensitivity and precision which are crucial for a clinical testing platform were investigated. Performance of NGS was found to be to be in-par or better and more informative as compared to the classical STR platform. Accurate performance was also evident by complete acceptability of on the ASHI EMO survey.
Author Response
Manuscript ID: IJMS-2503721
Thank you for giving us the opportunity to submit a revised draft of the manuscript “Chimerism testing by Next Generation Sequencing for detection of engraftment and early disease relapse in allogeneic hematopoietic cell transplantation and an overview of NGS chimerism studies” for publication in the ‘International Journal of Molecular Sciences’.
We appreciate the time and effort that you and the reviewers dedicated to providing feedback on our manuscript and we appreciate the insightful comments and helpful suggestions for improving our paper. We have incorporated all the suggestions made by the reviewers. Those changes are highlighted within the revised manuscript. Please see below, highlighted with the blue reply, for a point-by-point response to the reviewers’ comments and concerns.
Comments and Suggestions for Authors
Reviewer 1
The study compares the applicability of NGS-based chimerism monitoring to the gold-standard STR-PCR assay. Several performance parameters like accuracy, sensitivity and precision which are crucial for a clinical testing platform were investigated. Performance of NGS was found to be to be in-par or better and more informative as compared to the classical STR platform. Accurate performance was also evident by complete acceptability of on the ASHI EMO survey.
Author's Reply to the Review Report (Reviewer 1):
Thank you very much for the appreciation.
Reviewer 1 comment: “Accurate performance was also evident by complete acceptability of on the ASHI EMO survey”
Reply: Yes, thank you very much for pointing out the survey results. The ASHI engraftment (EMO) survey is designed to evaluate a laboratory’s ability to determine the proportion of DNA or cells from two individuals in a mixture of their blood samples, as a model of testing for hematologic chimerism. We are part of the ASHI EMO program which included five unknown chimeric mixtures per survey, we participated in two surveys which comprising a total of 10 unknown mixtures. We have received ‘GOOD’ grade (100%) in all surveys. The results from this program demonstrate a high level of assay accuracy and quality.
Reviewer 2 Report
Comments and Suggestions for Authors
The article entitled “Chimerism testing by Next Generation Sequencing for detection of engraftment and early disease relapse in allogeneic hematopoietic cell transplantation and an overview of NGS chimerism studies” focuses on an clinical important topic such as the allogenic hematopoietic cell transplantation (allo-HCT), the only curative treatment for patients with hematologic neoplasms, in reference to monitoring patient/donor cell ratios called “chimerism” using different molecular biology assays such as Next Generation Sequencing (NGS) and Short Tandem Repeat (STR). In addition, the authors conducted a review about other methods such as qPCR, STR-PCR, VNTRs, SNP. The results of this study showed that NGS has a superiority in terms of sensitivity, precision and quantitative accuracy. The study purpose is appropriate. In relation to material and methods, the authors evaluated 174 DNA samples from 59 individuals, but they do not report the diagnosis of the disease. This information would be reported (hematologic neoplasm or not hematologic neoplasm). In addition the authors state that this study validates NGS against the others but in paragraph “Conclusions” state that “More research is required to validate the NGS based method….”. Therefore, I think that this declaration has to be deleted. I think that this article is not suitable for publication in its current version.
Author Response
Manuscript ID: IJMS-2503721
Thank you for giving us the opportunity to submit a revised draft of the manuscript “Chimerism testing by Next Generation Sequencing for detection of engraftment and early disease relapse in allogeneic hematopoietic cell transplantation and an overview of NGS chimerism studies” for publication in the ‘International Journal of Molecular Sciences’.
We appreciate the time and effort that you and the reviewers dedicated to providing feedback on our manuscript and we appreciate the insightful comments and helpful suggestions for improving our paper. We have incorporated all the suggestions made by the reviewers. Those changes are highlighted within the revised manuscript. Please see below, highlighted with the blue reply, for a point-by-point response to the reviewers’ comments and concerns.
Author's Reply to the Review Report (Reviewer 2):
Comments and Suggestions for Authors
The article entitled “Chimerism testing by Next Generation Sequencing for detection of engraftment and early disease relapse in allogeneic hematopoietic cell transplantation and an overview of NGS chimerism studies” focuses on an clinical important topic such as the allogenic hematopoietic cell transplantation (allo-HCT), the only curative treatment for patients with hematologic neoplasms, in reference to monitoring patient/donor cell ratios called “chimerism” using different molecular biology assays such as Next Generation Sequencing (NGS) and Short Tandem Repeat (STR).
In addition, the authors conducted a review about other methods such as qPCR, STR-PCR, VNTRs, SNP. The results of this study showed that NGS has a superiority in terms of sensitivity, precision and quantitative accuracy.
Comment 1: The study purpose is appropriate.
Answer 1: Thank you very much, we appreciate the response.
Comment 2: In relation to material and methods, the authors evaluated 174 DNA samples from 59 individuals, but they do not report the diagnosis of the disease. This information would be reported (hematologic neoplasm or not hematologic neoplasm).
Answer 2: Thank you very much for this suggestion. We have corrected the text. The total number of DNA samples included in this study was 174, from 46 individuals and 10 ASHI External proficiency EMO samples. In supplementary figure we have shown ASHI EMO samples and therefore total number was 184 (174+10=184). We have corrected the number throughout in the revised manuscript, including in the Tables and figures.
Further, as suggested by the reviewer, we are incorporating a Table 2 from 46 individuals with hematological and non-hematological malignancies in revised manuscript. In the material method section, we have added a text which could be read as “Patients enrolled in the validation study were divided according to their disease diagnoses. We had the disease status for forty-three patients, out of which, 55.8% of patients were diagnosed with Acute Myeloid Leukemia (AML=24), 16.3% Myelodysplastic Syndromes (MDS=7), and 7% of each of Acute Lymphocytic Leukemia (ALL=3) and Hodgkins Lymphoma (HL=3). Two patients were diagnosed with Chronic Myelomonocytic Leukemia (CMML=2; 4.7%) and one each of Chronic Myelogenous Leukemia (CML=1; 2.3%), Neuroblastoma (NBL=1; 2.3%), Severe combined immunodeficiency (SCID=1; 2.3%) and Telomere Biology Disorder (TBD=1; 2.3%) (Table 2), “Materials and Methods section”, Sample selection, Line number 200-208.
|
Table 2: Distribution of patient disease diagnoses included for validation study |
||
|
Disease Diagnosis |
N=46 |
% |
|
Acute Lymphocytic Leukemia (ALL) |
3 |
7.0 |
|
Acute Myeloid Leukemia (AML) |
24 |
55.8 |
|
Chronic Myelogenous Leukemia (CML) |
1 |
2.3 |
|
Chronic Myelomonocytic Leukemia (CMML) |
2 |
4.7 |
|
Hodgkins Lymphoma (HL) |
3 |
7.0 |
|
Myelodysplastic Syndromes (MDS) |
7 |
16.3 |
|
Neuroblastoma (NBL) |
1 |
2.3 |
|
Severe combined immunodeficiency (SCID) |
1 |
2.3 |
|
Telomere Biology Disorder (TBD) |
1 |
2.3 |
|
Unknown |
3 |
7.0 |
Comment 3: In addition the authors state that this study validates NGS against the others but in paragraph “Conclusions” state that “More research is required to validate the NGS based method….”. Therefore, I think that this declaration has to be deleted.
Answer 3: We agree with the reviewer’s assessment and as suggested, we have deleted the statement from the conclusion ‘More research is required to validate the NGS based method for chimerism monitoring prior to its implementation in clinical laboratories serving diverse ethnic population’. ‘Conclusions’, line number 453 -455 deleted.
Reviewer 3 Report
Comments and Suggestions for Authors
In this manuscript, the authors evaluate the AlloSeq HCT NGS assay for chimerism monitoring after allogeneic hematopoietic stem cell transplantation (allo-HCT) and compare it with the most commonly used STR-PCR assay. The study includes a significant number of samples compared to most other published studies comparing NGS-based assays to so far validated and widely used techniques for chimerism assessment. There are though two major concerns that rise: a) the novelty of the study and, b) some inaccuracies noticed and the general presentation of the manuscript, especially the disorganized presentation of figures, and discordances when they are referred to in the text, making it hard for the reader to follow (Regarding the second one, relevant issues are analyzed below and should be addressed by authors before submitting a manuscript). It is though interesting that the authors review the relevant literature trying to make comparisons with the presented research, thus possibly unraveling its advantages.
Comment 1
There are inaccuracies in the number of samples described in different parts of the manuscript. Namely, in the abstract the authors mention 174 specimens, while in the supplementary figure provided there seem to be 184 and in table 1 the sample size appears to be 150. The number of individuals tested appears to be 59 in the abstract, while in the materials and methods section 3.1 the authors mention “(18) for donor and recipient in pre-transplant samples” in an incoherent way.
Comment 2
Figure 4 is irrelevantly placed in the text. Namely, figure 4a is metntioned in section 2.1 and shown after section 2.6, while Figure 4b is mentioned in section 2.2 ad shown after 2.6. The authors should consider replacing figure 4 after section 2.2.
Comment 3
Figure 4a: There are only 5 samples presented, while 10 samples were tested. If the authors chose to include representative samples in the figure, they should mention it in the figure legend.
Comment 4
Figure 1: The legends or the figure parts 1a and 1b are possibly placed inversely. Make sure the parts are also mentioned correctly in the text because it is quite hard for the reader to follow. It is recommended to define what series 1 and 2 are. For figure 1b: it is quite difficult to decipher the different linear representations. The authors should consider a clearer way of illustration.
Comment 5
Lines 119-120: “In addition, NGS produces additional number of informative markers in the same samples where STR-PCR has a limited number of informative markers (n=110, figure 4b)”. In figure 4b, only NGS informative markers are presented. There is no information abour STR-PCR informative markers in the graph to illustrate the comparison.
Comment 6
The figures mentioned in the manuscript section 2.4 are probably figure 2 parts (not 3).
Comment 7
The subject of this study is not really novel, as proven by the other published studies comparing NGS to STR-PCR, which are indeed reviewed by the authors in an effort to present useful conclusions via comparison. It is recognized that the number of samples studied in this study deserves to be noted, but one can argue that the study of Matthew F. et al. presented in Table 1 includes comparable sample size and number of SNPs used as markers. If limited to the studies presented in Table 1, we could highlight that the presented study is the only one testing the AlloSep HCT NGS assay in the setting of chimerism assessment post-allo-HSCT. Nevertheless, the authors do not mention the recent study of Picard C. et al. (Front Immunol. 2023; 14: 1023116. doi: 10.3389/fimmu.2023.1023116), where AlloSeq HCT is compared to other chimerism quantification methods, namely two new digital PCR systems [QIAcuity dPCR (Qiagen) and QuantStudio Absolute Q (ThermoFisher)] and another NGS-based assay [NGStrack (GenDX)], and which the advantages of the effectiveness, as well as the advantages of the method are analyzed.
Comments on the Quality of English LanguageMinor editing is required.
Author Response
Manuscript ID: IJMS-2503721
Thank you for giving us the opportunity to submit a revised draft of the manuscript “Chimerism testing by Next Generation Sequencing for detection of engraftment and early disease relapse in allogeneic hematopoietic cell transplantation and an overview of NGS chimerism studies” for publication in the ‘International Journal of Molecular Sciences’.
We appreciate the time and effort that you and the reviewers dedicated to providing feedback on our manuscript and we appreciate the insightful comments and helpful suggestions for improving our paper. We have incorporated all the suggestions made by the reviewers. Those changes are highlighted with the blue color text within the revised manuscript. Please see below, highlighted with the blue reply, for a point-by-point response to the reviewers’ comments and concerns.
Author's Reply to the Review Report (Reviewer 3):
Comments and Suggestions for Authors
In this manuscript, the authors evaluate the AlloSeq HCT NGS assay for chimerism monitoring after allogeneic hematopoietic stem cell transplantation (allo-HCT) and compare it with the most commonly used STR-PCR assay. The study includes a significant number of samples compared to most other published studies comparing NGS-based assays to so far validated and widely used techniques for chimerism assessment. There are though two major concerns that rise: a) the novelty of the study and, b) some inaccuracies noticed and the general presentation of the manuscript, especially the disorganized presentation of figures, and discordances when they are referred to in the text, making it hard for the reader to follow (Regarding the second one, relevant issues are analyzed below and should be addressed by authors before submitting a manuscript). It is though interesting that the authors review the relevant literature trying to make comparisons with the presented research, thus possibly unraveling its advantages.
Comment 1
There are inaccuracies in the number of samples described in different parts of the manuscript. Namely, in the abstract the authors mention 174 specimens, while in the supplementary figure provided there seem to be 184 and in table 1 the sample size appears to be 150. The number of individuals tested appears to be 59 in the abstract, while in the materials and methods section 3.1 the authors mention “(18) for donor and recipient in pre-transplant samples” in an incoherent way.
Answer 1: The total number of samples included in this study was 174, and 10 ASHI External proficiency EMO samples. In supplementary figure we have shown ASHI EMO samples and therefore total number was 184 (174+10=184). We have corrected the number throughout in the revised manuscript, including in the Tables and figures.
Sub-comment 1.1: while in the materials and methods section 3.1 the authors mention “(18) for donor and recipient in pre-transplant samples” in an incoherent way.
Answer 1.1: Here 18 is the reference number (ref 18) and not the number of samples. To avoid any confusion, we have edited the test which could be read as “The criteria for choosing the STR-PCR chimerism samples for comparison with NGS were dependent upon the presence of one or more informative markers. When recipient and donor have no common allele, which is fully informative, or when recipient and donor share one common allele, which is heterozygous, but still informative (18)”. Materials and Methods, 3.1. Sample selection, line number 207-209.
Comment 2
Figure 4 is irrelevantly placed in the text. Namely, figure 4a is metntioned in section 2.1 and shown after section 2.6, while Figure 4b is mentioned in section 2.2 and shown after 2.6. The authors should consider replacing figure 4 after section 2.2.
Answer 2: We are agree with the reviewer, we have edited the figure number 4 as a figure number 2 (fig 4à in now fig 2) and likewise we have changed all figure numbers as well as the text in the revised manuscript wherever it was described.
Comment 3
Figure 4a: There are only 5 samples presented, while 10 samples were tested. If the authors chose to include representative samples in the figure, they should mention it in the figure legend.
Answer 3: Figure 4a (which is now figure2): As suggested by the reviewer, we have changed the figure legend as “2a: Results for testing 5 representative ASHI 2022 EMO-1 survey (test samples EMO-156, EMO-157, EMO-158, EMO-159, EMO-160)”. And in the text “Results for testing 5 representative ASHI 2022 EMO-1 survey (test samples EMO-156, EMO-157, EMO-158, EMO-159, EMO-160 are shown in figure 2a which demonstrate a high level of assay accuracy and quality (Figure 2a)”, Results section, 2.1 Accuracy and proficiency testing, line number 100-103.
Comment 4
Figure 1: The legends or the figure parts 1a and 1b are possibly placed inversely. Make sure the parts are also mentioned correctly in the text because it is quite hard for the reader to follow. It is recommended to define what series 1 and 2 are. For figure 1b: it is quite difficult to decipher the different linear representations. The authors should consider a clearer way of illustration.
Answer 4: By carefully reading the figure 1 legends, we realized that there were a few text copied and paste to keep the formatting same, which we have deleted. However, to make it more clear, we have edited the heading and have updated in the figure and revised manuscript. The revised edited text in figure could be read as “Figure 1: NGS concordance in the Known (artificial mixed chimerism) and NGS concordance with STR in blind samples. 1a. Results of artificial mixed chimerism with two known donor DNA samples at varied level of simulated artificial mixed chimerism. The results represent good correlation of the expected chimerism in percent DNA fractions in samples with two unrelated genetic contributors. Signal linearity with the expected % DNA fraction was R2>0.99 across all donors. 1b: Comparison of NGS concordance with STR in blind samples received from Children Hospital of Philadelphia. The results showed a high correlation (R2>0.99) between expected (STR) and observed chimerism levels (NGS)”.
Comment 4a: It is recommended to define what series 1 and 2 are. For figure 1b: it is quite difficult to decipher the different linear representations. The authors should consider a clearer way of illustration.
Answer 4a: It was a blank column which was selected while performing the analysis which is deleted now in the revised figure and manuscript. We have deleted the extra lines and unwanted text from the figure 1a and 1b. We have enhanced the quality of all figure and have used bigger font size to make better illustration in the revised manuscript.
Comment 5
Lines 119-120: “In addition, NGS produces additional number of informative markers in the same samples where STR-PCR has a limited number of informative markers (n=110, figure 4b)”. In figure 4b, only NGS informative markers are presented. There is no information abour STR-PCR informative markers in the graph to illustrate the comparison.
Answer 5: The comparison of informative marker in PCR-STR vs NGS has been shown in the table 1 (last reference, current study). We wanted to show only the average number NGS informative markers in the figure and therefore we have edited the text which could be read as “In addition, NGS produces additional number of informative markers in the same samples where STR-PCR has a limited number of informative markers (table 1). The average number of informative markers using NGS AlloSeq HCT was 110 (Figure 2b)”. Section ‘Comparison of NGS chimerism with STR-PCR’, Line numbers 125-128.
Comment 6
The figures mentioned in the manuscript section 2.4 are probably figure 2 parts (not 3).
Answer 6: We have carefully checked all figures and as suggested above (comment 2), we have changed all figures and their number as well as the text in the revised manuscript wherever it was described.
Comment 7
The subject of this study is not really novel, as proven by the other published studies comparing NGS to STR-PCR, which are indeed reviewed by the authors in an effort to present useful conclusions via comparison. It is recognized that the number of samples studied in this study deserves to be noted, but one can argue that the study of Matthew F. et al. presented in Table 1 includes comparable sample size and number of SNPs used as markers. If limited to the studies presented in Table 1, we could highlight that the presented study is the only one testing the AlloSep HCT NGS assay in the setting of chimerism assessment post-allo-HSCT. Nevertheless, the authors do not mention the recent study of Picard C. et al. (Front Immunol. 2023; 14: 1023116. doi: 10.3389/fimmu.2023.1023116), where AlloSeq HCT is compared to other chimerism quantification methods, namely two new digital PCR systems [QIAcuity dPCR (Qiagen) and QuantStudio Absolute Q (ThermoFisher)] and another NGS-based assay [NGStrack (GenDX)], and which the advantages of the effectiveness, as well as the advantages of the method are analyzed.
Answer 7: Thank you very much for the valuable suggestions. We appreciate the response and as suggested to incorporate the line we have added the line in the revised manuscript in the ‘Discussion’ section line number 361-363. The incorporated lines could be read as “Further, to the best of our knowledge, the presented study is the only one testing the AlloSeq HCT NGS assay in the setting of chimerism assessment post-allo-HCT from the USA, Pennsylvania, Philadelphia”.
Also, we have addressed the paper published by Pcard et al. When we communicate this paper into the group to the co-authors for review in Jan-2023, the paper by Picard et al was not published online and we missed this paper to include in the discussion. However, as suggested by the reviewer, we have included the paper Picard et al in the revised manuscript. Picard et al. have general comparison between all the techniques (Table 1. Comparison of STR, qPCR, dPCR and NGS technologies) and between the three NGS Kits (Devyser Chimerism NGS vs AlloSeq HCT vs NGStrack). However, in the present manuscript, we have compared and included all those studies who has NGS as a primary method for chimerism detection and STR or ddPCR technique as a gold standard for comparison.
We have added the following text in the ‘Discussion’ section, line numbers 408-421. The added para could be read as “Recently, a paper publication published by Picard et al (44) using a total of 38 samples, a comparison of chimerism quantification data for two new digital PCR systems and two NGS-based chimerism quantification methods were performed. They found that all the three existing NGS kits, Devysr® (Devyser Chimerism), CareDX (AlloSeq HCT) and GenDx (NGStrack) are similar in terms of analytical performance. They concluded that AlloSeq proposes an analysis in the absence of a contributor and it is not necessary to perform genotyping for all contributors. In case of AlloSeq HCT due to presence of the high number of markers, it avoids false-negative results caused by chromosomal deletions in the relapse of some malignancies. Chimerism detection using NGS AlloSeq HCT kit is the easiest technique, with only one mix (vs. 7 for NGStrack) and one PCR (vs. 2 for Devyser chimerism NGS). However, they also concluded that in comparison of all new methods in clinical practice, digital PCR is faster (result within the day vs. 72 hours), easier to use and easier to interpret than that of all NGS methods, which enable earlier detection of relapses in allo-HCT”.
In addition, we have edited the Materials and Methods section, 3.4. NGS, panel and chimerism analysis, line number 243-251 “The AlloSeq HCT assay is a targeted next generation sequencing assay that makes use of variations in single nucleotide polymorphisms (SNPs) to calculate the proportion of recip-ient and donor-derived DNA which is present in a post-transplant sample. The sequenc-ing reaction was carried out using the MiSeq v3 Reagent kit for 150 cycles. For the se-quencing reaction, a customized sample sheet was created, and AlloSeq HCT Software was used to analyze the data from the fastq files. Amplification and indexing are combined in-to a single reaction by the special amplification procedure, which reduces human interac-tion and manipulation errors. For all the PCR programs and step by step procedure, we followed the instruction manual protocol”. And Line number 258-270 “Sequence data is analyzed using AlloSeq HCT software which outputs the percentage of DNA (% DNA) for up to three distinct genomes (genetic contributors) detected in post-transplant samples. The calculation of % DNA from recipient and donor(s) present in post-transplant samples is achieved by determining the fraction of different nucleotides sequenced at each SNP location evaluated. Recipient and donor(s) genotypes obtained using AlloSeq HCT are required to calculate the % DNA obtained from each genetic con-tributor present in the post-transplant sample. The number of reads categorized as "refer-ence" (Ref) and "alternative" (Alt) of the Reference Sequence hg19 (human genome 19; Ge-nome Reference Consortium) is used to calculate the variant allele frequency (VAF) for each marker. The analysis time for each sample is 30 seconds, and multiple samples can be analyzed on AlloSeq HCT software simultaneously. The software is user friendly. It keeps track of the genotyping data it automatically detects if there is contamination or mix-ups between the samples”.
Thank you

Reviewer 4 Report
Comments and Suggestions for Authors
This study aims to establish and detect mixed chimerism and minimal residual disease using NGS testing for the evaluation of engraftment and to detect early relapse after allo-HCT. Meanwhile, this study also try to compare the data with the existing laboratory method based on STR analysis. Totally 174 DNA specimens from 59 individuals were assessed using a commercially available kit for NGS, AlloSeq HCT NGS (CareDx) and the STR-PCR assay. A significant correlation was observed between NGS and STR-PCR chimerism methods. High accuracy was attained for the external proficiency testing program for two surveys. The authors concluded that NGS based chimerism analysis using the AlloSeq HCT assay is a reliable and accurate method for donor-recipient chimerism quantification, early relapse monitoring, and minimal residual disease in patients.
1) The overall writing has some formatting issues, like wording and spacing. I suggest the authors check the grammar and avoid any typos. More importantly, the writing needs improvement.
2) The method part is lack of details. More detailed descriptions are needed to explain this part.
3) For almost all figures shown in this manuscript, the font size is small, and the resolution is very poor. I would recommend the authors to polish the figures with high resolution and larger font size.
4) The results are not quite sufficient. More discussions on the result part are needed. Moreover, I would suggest the authors discuss using different approaches (e.g., PMID: 36545790; PMID: 35284940) as future perspectives, to facilitate the donor-recipient chimerism quantification and early relapse monitoring of minimal residual disease.
Comments on the Quality of English LanguageThe overall writing has some formatting issues, like wording and spacing. I suggest the authors check the grammar and avoid any typos. More importantly, the writing needs improvement.
Author Response
Manuscript ID: IJMS-2503721
Thank you for giving us the opportunity to submit a revised draft of the manuscript “Chimerism testing by Next Generation Sequencing for detection of engraftment and early disease relapse in allogeneic hematopoietic cell transplantation and an overview of NGS chimerism studies” for publication in the ‘International Journal of Molecular Sciences’.
We appreciate the time and effort that you and the reviewers dedicated to providing feedback on our manuscript and we appreciate the insightful comments and helpful suggestions for improving our paper. We have incorporated all the suggestions made by the reviewers. Those changes are highlighted with the blue color within the revised manuscript. Please see below, highlighted with the blue reply, for a point-by-point response to the reviewers’ comments and concerns.
Author's Reply to the Review Report (Reviewer 4):
Comments and Suggestions for Authors
This study aims to establish and detect mixed chimerism and minimal residual disease using NGS testing for the evaluation of engraftment and to detect early relapse after allo-HCT. Meanwhile, this study also try to compare the data with the existing laboratory method based on STR analysis. Totally 174 DNA specimens from 59 individuals were assessed using a commercially available kit for NGS, AlloSeq HCT NGS (CareDx) and the STR-PCR assay. A significant correlation was observed between NGS and STR-PCR chimerism methods. High accuracy was attained for the external proficiency testing program for two surveys. The authors concluded that NGS based chimerism analysis using the AlloSeq HCT assay is a reliable and accurate method for donor-recipient chimerism quantification, early relapse monitoring, and minimal residual disease in patients.
- The overall writing has some formatting issues, like wording and spacing. I suggest the authors check the grammar and avoid any typos. More importantly, the writing needs improvement.
Answer 1: We have taken care of the formatting, grammar, typos and English language and the manuscript was further reviewed by all authors including English Vetted author. We have edited the manuscript for English language and have highlighted the edited part with the blue color text.
- The method part is lack of details. More detailed descriptions are needed to explain this part.
Answer 2: Thank you for this suggestion and as suggested by the reviewer, the method section have been elaborated and edited. Further, we are incorporating a Table 2 from 46 individuals with hematological disorders in revised manuscript. In the material method section, we have added a text which could be read as “Patients enrolled in the validation study were divided according to their disease diagnoses. We had the disease status for forty-three patients, out of which, 55.8% of patients were diagnosed with Acute Myeloid Leukemia (AML=24), 16.3% Myelodysplastic Syndromes (MDS=7), and 7% of each of Acute Lymphocytic Leukemia (ALL=3) and Hodgkins Lymphoma (HL=3). Two patients were diagnosed with Chronic Myelomonocytic Leukemia (CMML=2; 4.7%) and one each of Chronic Myelogenous Leukemia (CML=1; 2.3%), Neuroblastoma (NBL=1; 2.3%), Severe combined immunodeficiency (SCID=1; 2.3%) and Telomere Biology Disorder (TBD=1; 2.3%) (Table 2), “Materials and Methods” section, 3.1. Sample selection, Line number 200-211.
|
Table 2: Distribution of patient disease diagnoses included for validation study |
||
|
Disease Diagnosis |
N=46 |
% |
|
Acute Lymphocytic Leukemia (ALL) |
3 |
7.0 |
|
Acute Myeloid Leukemia (AML) |
24 |
55.8 |
|
Chronic Myelogenous Leukemia (CML) |
1 |
2.3 |
|
Chronic Myelomonocytic Leukemia (CMML) |
2 |
4.7 |
|
Hodgkins Lymphoma (HL) |
3 |
7.0 |
|
Myelodysplastic Syndromes (MDS) |
7 |
16.3 |
|
Neuroblastoma (NBL) |
1 |
2.3 |
|
Severe combined immunodeficiency (SCID) |
1 |
2.3 |
|
Telomere Biology Disorder (TBD) |
1 |
2.3 |
|
Unknown |
3 |
7.0 |
In addition, we have added two paragraphs in the 3.4 NGS, panel and chimerism analysis line number 243-251, which could be read as “The AlloSeq HCT assay is a targeted next generation sequencing assay that makes use of variations in single nucleotide polymorphisms (SNPs) to calculate the proportion of recip-ient and donor-derived DNA which is present in a post-transplant sample. The sequenc-ing reaction was carried out using the MiSeq v3 Reagent kit for 150 cycles. For the se-quencing reaction, a customized sample sheet was created, and AlloSeq HCT Software was used to analyze the data from the fastq files. Amplification and indexing are combined in-to a single reaction by the special amplification procedure, which reduces human interac-tion and manipulation errors. For all the PCR programs and step by step procedure, we followed the instruction manual protocol”.
And line number 258-270, “Sequence data is analyzed using AlloSeq HCT software which outputs the percentage of DNA (% DNA) for up to three distinct genomes (genetic contributors) detected in post-transplant samples. The calculation of % DNA from recipient and donor(s) present in post-transplant samples is achieved by determining the fraction of different nucleotides sequenced at each SNP location evaluated. Recipient and donor(s) genotypes obtained using AlloSeq HCT are required to calculate the % DNA obtained from each genetic con-tributor present in the post-transplant sample. The number of reads categorized as "refer-ence" (Ref) and "alternative" (Alt) of the Reference Sequence hg19 (human genome 19; Ge-nome Reference Consortium) is used to calculate the variant allele frequency (VAF) for each marker. The analysis time for each sample is 30 seconds, and multiple samples can be analyzed on AlloSeq HCT software simultaneously. The software is user friendly. It keeps track of the genotyping data it automatically detects if there is contamination or mix-ups between the samples”.
- For almost all figures shown in this manuscript, the font size is small, and the resolution is very poor. I would recommend the authors to polish the figures with high resolution and larger font size.
Answer 3: We are agreeing with the reviewer; we have edited all figures. We have deleted the extra lines and unwanted text from all figures. We have enhanced the quality of figures and have used bigger font size to make better illustration in the revised manuscript. Considering the text we needed to change the figure numbers too, like fig number 4 as a figure number 2 (fig 4à in now fig 2). Likewise, we have changed all figures number as well as the text in the revised manuscript wherever it was described.
- The results are not quite sufficient. More discussions on the result part are needed. Moreover, I would suggest the authors discuss using different approaches (e.g., PMID: 36545790; PMID: 35284940) as future perspectives, to facilitate the donor-recipient chimerism quantification and early relapse monitoring of minimal residual disease.
Answer 4: As suggested, we have incorporated a new table and a para in the Materials and Methods section, (3.1. Sample selection, line number 200-211) and in results section of the revised manuscript.
We have added two paragraph in the 3.4. NGS, panel and chimerism analysis, line number 243-251, which could be read as “The AlloSeq HCT assay is a targeted next generation sequencing assay that makes use of variations in single nucleotide polymorphisms (SNPs) to calculate the proportion of recip-ient and donor-derived DNA which is present in a post-transplant sample. The sequenc-ing reaction was carried out using the MiSeq v3 Reagent kit for 150 cycles. For the se-quencing reaction, a customized sample sheet was created, and AlloSeq HCT Software was used to analyze the data from the fastq files. Amplification and indexing are combined in-to a single reaction by the special amplification procedure, which reduces human interac-tion and manipulation errors. For all the PCR programs and step by step procedure, we followed the instruction manual protocol” and line number 258-270 “Sequence data is analyzed using AlloSeq HCT software which outputs the percentage of DNA (% DNA) for up to three distinct genomes (genetic contributors) detected in post-transplant samples. The calculation of % DNA from recipient and donor(s) present in post-transplant samples is achieved by determining the fraction of different nucleotides sequenced at each SNP location evaluated. Recipient and donor(s) genotypes obtained using AlloSeq HCT are required to calculate the % DNA obtained from each genetic con-tributor present in the post-transplant sample. The number of reads categorized as "refer-ence" (Ref) and "alternative" (Alt) of the Reference Sequence hg19 (human genome 19; Ge-nome Reference Consortium) is used to calculate the variant allele frequency (VAF) for each marker. The analysis time for each sample is 30 seconds, and multiple samples can be analyzed on AlloSeq HCT software simultaneously. The software is user friendly. It keeps track of the genotyping data it automatically detects if there is contamination or mix-ups between the samples”.
This is the validation study and has comparison with the other studies published online. However, as suggested by the reviewer we have added a paragraph of future prospective in the discussion section. The added paragraph could be read as “There are other approaches, to facilitate the donor-recipient chimerism quantification and early relapse monitoring of minimal residual disease, recent developments in single cell spatial transcriptomic technology enable the examination of the spatial patterns of cell communication and hold promise for unraveling the intricate ligand-receptor interactions that occur across different cell types (45). Triozzi et al. (46) suggested the use of blood bio-energetics and metabolomics as predictive biomarkers of patient response to immune checkpoint inhibitor therapy.”. “Discussion” section Line number 430-444.
We have added one more study (reference#44) in the Table Number-1 and have discussed in the ‘Discussion’ section, Line number 409-422, the added text could be read as “Recently, a paper publication published by Picard et al (44) using a total of 38 sam-ples, a comparison of chimerism quantification data for two new digital PCR systems and two NGS-based chimerism quantification methods were performed. They found that all the three existing NGS kits, Devysr® (Devyser Chimerism), CareDX (AlloSeq HCT) and GenDx (NGStrack) are similar in terms of analytical performance. They concluded that AlloSeq proposes an analysis in the absence of a contributor and it is not necessary to perform genotyping for all contributors. In case of AlloSeq HCT due to presence of the high number of markers, it avoids false-negative results caused by chromosomal deletions in the relapse of some malignancies. Chimerism detection using NGS AlloSeq HCT kit is the easiest technique, with only one mix (vs. 7 for NGStrack) and one PCR (vs. 2 for Devyser chimerism NGS). However, they also concluded that in comparison of all new methods in clinical practice, digital PCR is faster (result within the day vs. 72 hours), easier to use and easier to interpret than that of all NGS methods, which enable earlier detection of relapses in allo-HCT”.
Thank you

Round 2
Reviewer 3 Report
Comments and Suggestions for Authors
Thank you for replying in detail to the comments made. All of them have been adequately addressed and the relevant revisions have been included in the text.
Reviewer 4 Report
Comments and Suggestions for Authors
I have no more concerns regarding this manuscript. It is ready to be accepted.